# Biological Evaluation of 8-Methoxy-2,5-dimethyl-5H-indolo[2,3-b] Quinoline as a Potential Antitumor Agent via PI3K/AKT/mTOR Signaling

**DOI:** 10.3390/ijms242015142

**Published:** 2023-10-13

**Authors:** Yunhao Ma, Hongmei Zhu, Xinrong Jiang, Zhongkun Zhou, Yong Zhou, Yanan Tian, Hao Zhang, Mengze Sun, Lixue Tu, Juan Lu, Yuqing Niu, Huanxiang Liu, Yingqian Liu, Peng Chen

**Affiliations:** 1School of Pharmacy, Lanzhou University, 199 Donggang West Road, Lanzhou 730000, China; myunhao2023@lzu.edu.cn (Y.M.); hmzhu@lzu.edu.cn (H.Z.); jiangxr20@lzu.edu.cn (X.J.); zhouzhk18@lzu.edu.cn (Z.Z.); zhouyong16@lzu.edu.cn (Y.Z.); zhanghao2020@lzu.edu.cn (H.Z.); sunmz20@lzu.edu.cn (M.S.); tulx21@lzu.edu.cn (L.T.); jlu21@lzu.edu.cn (J.L.); niuyq2021@lzu.edu.cn (Y.N.); 2Faculty of Applied Science, Macao Polytechnic University, Macao, China; p2214944@mpu.edu.mo (Y.T.); hxliu@mpu.edu.mo (H.L.)

**Keywords:** neocryptolepine derivatives, colorectal cancer, antiproliferative activity, apoptosis, PI3K/AKT/mTOR

## Abstract

Chemotherapy is commonly used clinically to treat colorectal cancer, but it is usually prone to drug resistance, so novel drugs need to be developed continuously to treat colorectal cancer. Neocryptolepine derivatives have attracted a lot of attention because of their good cytotoxic activity; however, cytotoxicity studies on colorectal cancer cells are scarce. In this study, the cytotoxicity of 8-methoxy-2,5-dimethyl-5H-indolo[2,3-b] quinoline (MMNC) in colorectal cells was evaluated. The results showed that MMNC inhibits the proliferation of HCT116 and Caco-2 cells, blocks the cell cycle in the G2/M phase, decreases the cell mitochondrial membrane potential and induces apoptosis. In addition, the results of western blot experiments suggest that MMNC exerts cytotoxicity by inhibiting the expression of PI3K/AKT/mTOR signaling pathway-related proteins. Based on these results, MMNC is a promising lead compound for anticancer activity in the treatment of human colorectal cancer.

## 1. Introduction

Colorectal cancer (CRC) is a common malignant tumor of the digestive system and represents a major therapeutic challenge [1,2]. According to the global cancer statistics in 2020, there are about 1.93 million new cases of colorectal cancer, and an estimated 0.93 million people had died of CRC worldwide, ranking 3rd in the incidence of malignant tumors and becoming the second leading cause of cancer death [3]. At the same time, the number of colorectal cancer cases and deaths in China is increasing year by year [4]. Unlike other cancers, such as lung cancer, there is no single risk factor for colorectal cancer. Epidemiological studies have identified that many factors have contributed to this increase, such as age and sex, inflammatory bowel disease, lifestyle or environmental changes [5]. Currently, most colorectal cancer patients are still treated with surgery and chemotherapy [6]. However, chemotherapy has the disadvantages of being highly toxic and prone to drug resistance, so novel drugs have to be constantly discovered to treat colorectal cancer [7].

Natural products are important source of drug discovery, As reported, 48.6% (85) of the 175 small-molecule antitumor agents approved from around the 1940s to 2012 are either natural products or analog-derived natural products [8,9,10]. Among numerous currently exploited medicinal African plant species, *Cryptolepis sanguinolenta* has recently received greater attention from an immense number of researchers and pharmaceutical companies [11]. Neocryptolepine is an indoquinoline alkaloid isolated from traditional African herbal medicine *Cryptolepis sanguinolenta*, and it has attracted a lot of attention because of its wide range of bioactivities [12]. Studies have reported that it has excellent cytotoxic activity against liver cancer, breast cancer and other cells. The structural modification of existing active compounds is a proven strategy to enhance activity. In our previous study, a series of neocryptolepine derivatives were obtained [12], of which compound 44 (MMNC) had excellent activity with an IC_50_ of 0.33 μM against colorectal cancer HCT116 cells.

The phosphatidylinositol 3-kinase (PI3K)/protein kinase B (AKT)/mammalian target of rapamycin (mTOR) signaling pathway is the most common activated signaling pathway in cancer including CRC [13]. It is one of the major signaling cascades that promotes cell survival and proliferation [14]. And it is becoming increasingly clear that inhibitors of PI3K/AKT/mTOR are effective in inhibiting tumor progression [15,16]. Apoptosis is a regulated cell death that plays an important role in the treatment of cancer cells. Inducing apoptosis is an important strategy for cancer treatment.

Alkaloids are a class of important plant-based anticancer drugs, which have great potential in the development of cancer therapeutics. Traditional alkaloids derived from plants have played a huge role in the past, and numerous studies have shown that alkaloids can often be used to treat a variety of cancers, including human breast cancer, liver cancer, oral cancer, colorectal cancer and stomach cancer [17]. Neocryptolepine has a wide range of biological activities: it can bind to DNA, it can inhibit topoisomerase II activity, and it has cytotoxicity, antibacterial, anti-malaria and other biological activities [11]. Studies have reported cytotoxic effects on various cancer cells, including lung cancer, cholangiocarcinoma, liver cancer, leukemia and ascites cancer cells [18,19,20]. It has been shown that some neocryptolepine derivatives have better anti-cancer cell proliferation activity after structural modification of neocryptolepine than that of the parent nucleus [21,22]. These studies have proved that neocryptolepine and its derivatives are very potential skeleton structures for novel drug discovery, and it is hoped that through further modification, analogues with better anti-tumor activity and higher safety factor can be obtained, or become novel lead compounds.

In this paper, we intend to verify the proapoptotic effect of MMNC and its regulation on PI3K/AKT/mTOR signaling pathway via co-incubation with human colorectal cancer HCT116 and Caco-2 cells. This work provides potential active lead compounds for the development of colorectal cancer drugs.

## 2. Results and Discussion

### 2.1. Chemistry

The synthetic route of MMN was obtained in our previous study [12]. The structures of target compound MMNC was confirmed via ^1^H NMR, ^13^C NMR, mass spectrometry and purity test, as shown in Appendix A.

### 2.2. MMNC Displays Potent Cytotoxic Ability in the Colorectal Cell Line

By evaluating the bioactivity of a series of neocryptolepine derivatives, we found that MMNC showed selective cytotoxicity to colorectal cancer HCT116 and Caco-2 cells, so MMNC was further tested and evaluated for anticancer activity. MMNC exhibited cytotoxicity against a variety of cancer cells (Table 1). As shown in Figure 1A, MMNC could inhibit the growth of HCT116 cells in a concentration-dependent and time-dependent manner. The IC_50_ of MMNC against HCT116 and Caco-2 cells was 0.33 μM and 0.51 μM, while 5-Fu, a CRC chemotherapy drug commonly used in clinical practice, had almost no inhibitory effect on HCT116 cells at the same concentration for 48 h (Figure 1B,D). Moreover, the IC_50_ of MMNC and 5-Fu in human intestinal epithelial HIEC cells was 11.27 μM and greater than 100 μM (Figure 1C). After MMNC treatment, the number of HCT116 and Caco-2 cells decreased significantly, and cells contracted obviously, had weak adhesion, and became round and loosely arranged (Figure 1E,G). When the concentration of MMNC was 0.1 μM and 0.2 μM, the number of HCT116 was still increasing. Therefore, at this concentration, MMNC did not directly cause the death of HCT116 cells, but inhibited cell growth (Figure 1H). In a word, MMNC displayed excellent cytotoxic activity in the colorectal cancer HCT116 and Caco-2 cells, and the cytotoxicity in human intestinal epithelial HIEC cells was low.

### 2.3. MMNC Inhibited the Colony Formation of HCT116 and Caco-2 Cells

Cell clonal formation experiment can directly reflect the proliferation ability of cells. To further verify the antitumor activity of MMNC in vitro, colony formation assay was performed. HCT116 and Caco-2 cells treated with different concentrations of MMNC (0, 0.1 and 0.2 μM) was cultured continuously for 7–10 days, and the cell cloning results of each group after staining are shown in Figure 2C,D. With the increase in MMNC concentration from 0 by 0.1 and 0.2 μM, the number of HCT116 and Caco-2 cell clones decreased in a concentration-dependent manner. These results suggested that MMNC could significantly inhibit tumor proliferation in vitro.

### 2.4. MMNC Arrested the HCT116 and Caco-2 Cells Cycle in the G2/M Stage

Cell-cycle interference by small molecules has been widely used to study cancer [23]. It has been reported that neocryptolepine intercalated into DNA and interfered with the catalytic activity of human topoisomerase II, provoking a massive accumulation of P388 murine leukemia cells in the G2/M phase [19]. To determine the mechanisms by which MMNC inhibits cell viability, flow cytometry was applied to examine cell cycle distribution. As shown in Figure 2A,B, cell cycle progression is significantly inhibited in the G2/M phase in a concentration-dependent manner. The percentage of G2/M phase cells increased from 12.6% to 62.7% and from 24.4% to 31.2% after treatment of different concentrations (0, 0.15 and 0.3 μM) of MMNC for 24 h. These results suggested that MMNC is a cell cycle-specific agent and induces cell cycle arrest at G2/M phase.

### 2.5. MMNC Can Induce Apoptosis of HCT116 Cells

To test the effect of MMNC on apoptosis of HCT116 and Caco-2 cells, we used Annexin V-Alexa Fluor 488/PI assay via flow cytometry. AV+/PI− represents early apoptosis cells, and AV+/PI+ represents late apoptosis cells. As shown in Figure 3A,B, MMNC mainly caused late apoptosis at all concentrations tested. The percentage of apoptotic cells (early and late stages) increased from 3.7% to 60.4% and from 7.8% to 57.0% as its concentration was increased from 0.15 to 0.6 μM in HCT116 and Caco-2 cells for 48 h, which demonstrated that MMNC significantly increased the cellular apoptosis in a concentration-dependent manner.

### 2.6. MMNC Reduced Cellular Mitochondrial Membrane Potential

The apoptosis pathway mainly includes endogenous pathway and exogenous death receptor pathway. The endogenous pathway is also known as the mitochondrial pathway. Moreover, the decrease in mitochondrial membrane potential is a landmark early event of apoptosis. Therefore, mitochondrial membrane potential experiment was performed on MMNC in HCT116 and Caco-2 cells to detect the changes in mitochondrial membrane potential. As shown in Figure 3C,D, with the MMNC concentration of treated cells increasing from 0 to 0.15, 0.3 μM and 0.6 μM, the mean proportion of cells with reduced mitochondrial membrane potential increased from 2.5% to 13.0%, 26.7% and 41.6% or from 19.2% to 33.7%, 40.0% and 64.6% in a concentration-dependent manner. The results of mitochondrial membrane potential experiments showed that MMNC could significantly reduce the membrane potential of HCT116 and Caco-2 cells in a concentration-dependent manner. This suggests that MMNC may induce apoptosis through mitochondrial pathway.

### 2.7. MMNC Can Promote the Production of Reactive Oxygen Species in HCT116 and Caco-2 Cells

Cancer cells can regulate the generation of reactive oxygen species, so that its content in cells can promote the growth and survival of cancer cells without causing apoptosis [24]. Therefore, inducing apoptosis by promoting ROS overproduction is also a cancer treatment strategy. In this study, whether MMNC affects the production of reactive oxygen species (ROS) in HCT116 and Caco-2 cells was studied, and the change in fluorescence intensity of DCF was detected via flow cytometry. After treating HCT116 and Caco-2 cells with different concentrations of MMNC (0, 0.15, 0.3 and 0.6 μM) for 48 h, the changes in ROS production in HCT116 and Caco-2 cells were detected. The results are as shown in Figure 4A,B: after HCT116 and Caco-2 cells were treated with different concentrations of MMNC, the mean DCF relative fluorescence intensity of HCT116 and Caco-2 cells increased from 1 to 1.18, 1.61 and 5.42 or from 1 to 1.34, 1.06 and 1.23 with the increase in MMNC concentration from 0 to 0.15 μM, 0.3 μM and 0.6 μM. The results of the reactive oxygen species experiment showed that MMNC promoted reactive oxygen species generation in HCT116 and Caco-2 cells.

### 2.8. MMNC Could Regulate the Expression of Proteins Related to Cells Cycle and the PI3K/AKT Cell Signaling Pathway in the HCT116 and Caco-2

In order to find the potential target of MMNC, molecular docking was employed. Specific scoring results are shown in Appendix A. Among them, MMNC had the best docking score with CDK1 protein, and the molecular docking results suggested that CDK1 might be the target of CMNC. Their interactions are shown in Figure 5A–C, with the imino H of the compound acting as a hydrogen bond donor forming a stable hydrogen bond interaction with the carbonyl oxygen atom on the 83 leucine (LEU) skeleton on the acceptor. Tryptophan 84 has strong polar interactions with ligand small molecules. Lysines 33 and 89 (LYS) are basic amino acids that are positively charged and interact with ligands in solvents. The results of molecular docking showed that CDK1 was a potential target of MMNC.

In addition, the PI3K/AKT/mTOR signaling pathway is one of the most frequently activated signaling pathways in colorectal cancer. In order to verify whether CDK1 is the target of MMNC, and whether MMNC exerts anti-proliferative activity by affecting the protein expression of PI3K/AKT/mTOR signaling pathway. In this study, Western blot experiments were performed to study the effects of MMNC on the protein and cell cycle-related expression levels of PI3K/AKT/mTOR signaling pathway. As shown in Figure 5D–G, after being treated with 0.3 μM MMNC for 48 h, the expressions of PI3KCA, p-AKT, m-TOR and p-mTOR proteins in HCT116 and Caco-2 cells were significantly down-regulated compared with the negative control group, while the content of AKT was not significantly changed in HCT116 cells. After treatment of HCT116 and Caco-2 cells with MMNC, the expression levels of CDK1 and mTOR proteins changed inconsistently. On the one hand, it may be that CDK1 and CyclinB1 protein can play a role in cell cycle regulation only after binding, and mTOR protein can only play an active role after phosphorylation. On the other hand, it may be due to the heterogeneity between tumor cells that results in this difference. The PI3K/AKT/mTOR signaling pathway is closely related to cell proliferation and growth, and the Western blot results suggested that MMNC may inhibit the proliferation of HCT116 and Caco-2 cells by inhibiting the continuous activation of PI3K/AKT/mTOR signaling pathway. In addition, the expression levels of CDK1 and Cyclin B1 proteins were also down-regulated. They are critical checkpoint proteins during the G2/M phase of the cell cycle, which explains how MMNC can induce cell cycle arrest in the G2/M phase. In summary, Western blot experiment results showed that MMNC could play an anti-proliferative role by inhibiting the activation of PI3K/AKT/mTOR pathway and the expression of cycle-related proteins in HCT116 and Caco-2 cells. Some studies have shown that neocryptolepine may exert cytotoxicity by inhibiting the activity of topoisomerase and inserting DNA [19]. Therefore, further experiments are needed to prove that MMNC may directly act on topoisomerase, thereby regulating the PI3K/AKT/mTOR cell signaling pathway and blocking cell cycle to exert cytotoxicity. Moreover, further experiments are needed to reveal the action of MMNC on targets in colorectal cancer cells and its interaction with related proteins on the PI3K/AKT/mTOR cell signaling pathway.

The PI3K (phosphatidylinositol-3 kinase)/AKT (protein kinase B)/mTOR (mammalian target of rapamycin) signaling pathway is a common and important cell signal transduction pathway that plays an important role in the progression of various cancers, regulating cell survival, growth, proliferation, angiogenesis and metabolism [15,25,26]. Studies have shown that in human cancers, including colorectal cancer, PIK3CA and PTEN, two important genes in the PI3K/AKT/mTOR pathway, are the most frequently mutated genes in this pathway [25,27,28]. Multiple genes in the PI3K/AKT/mTOR signaling pathway are frequently mutated in colorectal cancer, so this pathway gene is an important molecular therapeutic target in colorectal cancer treatment [29,30,31]. Therefore, if we can start with the effective strategy of targeting the inhibition of PI3K/AKT/mTOR signaling pathway and develop novel small-molecule inhibitors, it may provide a novel candidate drug for the clinical treatment of colorectal cancer.

## 3. Materials and Methods

### 3.1. Reagents and Synthesis

All synthetic materials were purchased from the reagent manufacturer, and no purification operations were carried out. All organic reagents were of analytical reagent grade. ^1^H NMR and ^13^C NMR spectra were recorded using a JNM-ECS-400 at 400 MHz. Mass spectrometry analysis was performed using a Bruker Micro TOF ESI-TOF mass spectrometer (Bruker Company, Billerica, MA, USA). All pH measurements during the experiment were performed using a PH-10C pH meter. MMNC was prepared as reported in our previous study [12].

### 3.2. Cell Culture

HCT116, Caco-2, AGS and SMMC-7721 cells were cultured in RPMI-1640 medium with 1% penicillin-streptomycin solution (PS) and 10% fetal bovine serum (FBS). PANC-1 and HIEC cells were cultured in Dulbecco’s modified Eagle’s medium with 1% penicillin-streptomycin solution (PS) and 10% fetal bovine serum (FBS). All cells were cultured in a CO_2_ incubator at 37 °C and 5% CO_2_.

### 3.3. Cytotoxicity Test

Cells were inoculated onto 96-well plates at a density of 1×10^4^ cells/well (100 μL) and incubated in an incubator for 24 h until the cells were plastered. The cells were then treated with different concentrations of MMNC (0, 0.1, 0.15, 0.2, 0.25, 0.3, 0.4 and 0.6 μM) for different times (24, 48 and 72 h), and 10 μL of MTT solution (5 mg/mL MTT, pH 7.2) was added to the cells and incubated for 4 h in the cell incubator. Then, the medium was discarded and 100 μL of dimethylsulfoxide was added to each well. The plates were gently shaken for 30 min and the absorbance of each well was measured at 490 nm using an enzyme marker. Based on the data from untreated control cells, the survival rate of treated cells was calculated using the following formula: Survival rate = 100% × (absorbance value of sample)/(absorbance value of control).

### 3.4. Colony Formation Assay

Cells (500 cells/well) were inoculated in 24-well cell culture plates and treated with different concentrations of MMNC (0, 0.1 and 0.2 μM) for 48 h. After 7 days of culture, cells were fixed with 4% paraformaldehyde for 1 h, then stained with 0.1% crystal violet for 30 min at room temperature, and finally, cells were washed twice with phosphate-buffered saline (PBS) and photographed for counting and analysis using Image J software (1.44, National Institutes of Health, Bethesda, MD, USA).

### 3.5. Analysis of Cell Cycle Arrest

HCT116 and Caco-2 cells were inoculated into six-well plates at 8 × 10^5^ cells per well and incubated for 24 h. After the cells were fully adhered to the wall, cells were treated with different concentrations of MMNC (0, 0.15, 0.3 μM) for 24 h. The cells were then digested and centrifuged and washed, after which the cells were fixed overnight in 75% ethanol. After overnight centrifugation and washing with phosphate-buffered saline (PBS), the cells were then treated with 100 μL RNase A for 30 min at 37 °C, followed by staining with 400 μL PI for 30 min. The final samples were put on the machine for determination using a flow cytometer (BD LSRFortessa, San Jose, CA, USA) and analyzed with Modifit software (3.2, Verity Software House Inc., Kirkland, MA, USA).

### 3.6. Analysis of Cell Apoptosis

HCT116 and Caco-2 cells were inoculated into six-well plates with 6 × 10^5^ cells per well and incubated for 24 h. After the cells were completely adhered to the wall, HCT116 and Caco-2 cells were treated with medium containing different concentrations of MMNC (0, 0.15, 0.3, 0.6 μM) for 48 h. Subsequently, the medium was collected and the cells were digested with EDTA-free trypsin, and all cells were collected by centrifugation and washed and centrifuged 3 times with PBS. The collected cells were then added to the binding buffer with 5 μL of Annexin V/Alexa Fluor 488 solution and incubated for 5 min at room temperature under light-proof conditions. The final samples were then added to the machine, assayed on a flow cytometer (BD LSRFortessa, San Jose, CA, USA) and analyzed with FlowJo software (10.4, Tree Star Inc., Ashland, OR, USA).

### 3.7. Mitochondrial Membrane Potential Analysis

HCT116 and Caco-2 cells were inoculated into six-well plates at 6 × 10^5^ cells per well and incubated for 24 h. After the cells were fully plastered, HCT116 and Caco-2 cells were treated with medium containing different concentrations of MMNC (0, 0.15, 0.3 and 0.6 μM) for 48 h. The cells were then digested and collected, then resuspended in 0.5 mL of 1640 medium, and 0.5 mL of 1 × JC-10 staining working solution was added and mixed thoroughly. The cells were incubated for 20 min at 37 °C in a cell incubator and mixed occasionally by inversion. During the incubation period, prepare an appropriate amount of 1 × JC-10 staining buffer and place in an ice bath. After centrifugation at the end of incubation, the cells were washed with 1 × JC-10 staining buffer and centrifuged again, followed by resuspension with 1 × JC-10 staining buffer and assayed using a flow cytometer (BD Biosciences, San Jose, CA, USA).

### 3.8. ROS Analysis

HCT116 and Caco-2 cells in the logarithmic growth stage were taken for digestion. HCT116 and Caco-2 cells were inoculated into the six-well plate with 6 × 10^5^ cells per well, and cultured for 24 h after the cells were completely attached to the wall. The HCT116 and Caco-2 cells were cultured in medium containing different concentrations of MMNC (0.15, 0.3, 0.6 μM) for 48 h. After 48 h, the medium was discarded, then the cells were digested and collected using trypsin without EDTA, washed with PBS and collected via centrifugation. The cells were then stained with a DCFH-DA probe for 20 min, during which the probe was shaken from time to time to make full contact with the cells, and then centrifuged to collect the cells, washed and centrifuged with PBS three times, and then suspended with PBS. Finally, LSRFortessa flow cytometer (BD) was used to detect the DCF fluorescence of cells, and the FlowJo software (Tree Star Inc., Ashland, OR, USA) package was used for data processing and analysis.

### 3.9. Molecular Docking Analysis

The molecular docking correlation analysis was performed using Schrodinger 10.1 software. The crystal structure of CDK1 protein binding to molecule in docking was obtained from PDB database (PDB:4Y72). The complex structure is first treated by removing crystal water, side chains and hydrogen atoms from the protein structure and minimizing the energy of the whole structure. The center of the small-molecule binding to the protein was used as the docking site of CFNC and the proteins for analysis. All structures of proteins can be obtained from PDB database. The PDB ID, resolution and organisms can be found in Appendix A.

### 3.10. Western Blotting Assay

HCT116 and Caco-2 cells were seeded in 6-well plates and cultured in complete medium until they reached confluence. Then, the cells were lysed in RIPA buffer containing protease inhibitor at 4 °C for 30 min. Total protein was resolved by SDS–PAGE and transferred to a PVDF membrane. The whole membrane was blocked with 5% dry nonfat milk in Tris-buffered saline plus 0.1% Tween for 2 h at room temperature. Proteins with different molecular weights were cut horizontally according to the position of marker bands, and then different protein bands were stripped and reprobed overnight with the corresponding primary antibody at 4 °C. Next, the membrane was incubated with the secondary HRP-conjugated antibody for 1 h at room temperature. Then, the membrane was cleaned with TBST, scanned using the BCL luminescence system for protein, and ImageJ for protein analysis. All experiments were performed at least three times. The average was used for data analysis.

### 3.11. Statistical Analysis

All the results were expressed as mean ± standard deviation, and the data were analyzed using SPSS 26.0 statistical software. Student’s t test (*t* test) was used to test the differences of analysis components, and *p* < 0.05 between groups indicated significant differences between the two groups.

## 4. Conclusions

In this study, we conducted the cytotoxicity of MMNC and evaluated the activity of MMNC in a variety of cancer cells. The MTT results showed that MMNC had good anti-proliferation activity after structural modification in several cancer cells. The IC_50_ of MMNC for HCT116, Caco-2, AGS, PANC-1 and SMMC-7721 was 0.33, 0.51 3.6, 18.4 and 9.7 μM. This also indicated that MMNC showed some selective cytotoxicity to colorectal cancer HCT116 and Caco-2 cells. Therefore, the anti-proliferation mechanism of CMNC on HCT116 and Caco-2 cells was further studied. The results of cell cloning further confirmed that MMNC could inhibit the proliferation of HCT116 and Caco-2 cells. The results of the cell cycle experiment showed that MMNC could induce cell cycle arrest in the G2/M phase. The results of the apoptosis experiment showed that MMNC could induce apoptosis. JC-10 mitochondrial membrane potential experiment showed that MMNC could reduce the mitochondrial membrane potential. The experimental results of reactive oxygen species generation showed that MMNC could promote the generation of reactive oxygen species. The molecular docking analysis showed that CDK1 was a potential target of MMNC. The results of the Western blot showed that MMNC may play an anti-proliferative role by inhibiting the activation of PI3K/AKT/mTOR pathway and the expression of cycle-related proteins in HCT116 and Caco-2 cells. These findings suggest that MMNC may be a potentially active lead compound for the treatment of colorectal cancer.

## Figures and Tables

**Figure 1 ijms-24-15142-f001:**
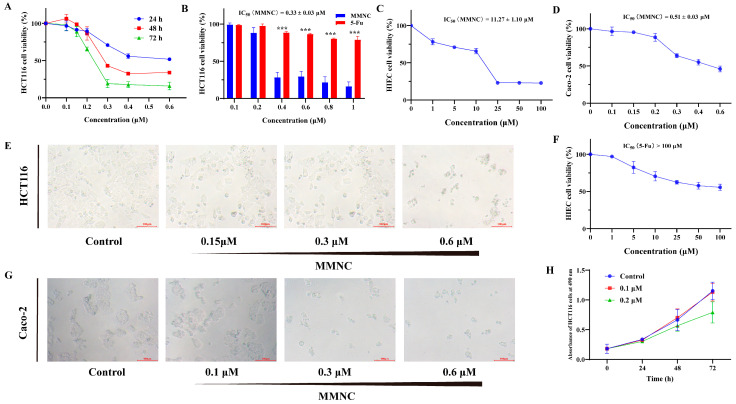
Cytotoxic effects of MMNC in different cells. (**A**) Cytotoxic effects of MMNC in HCT116 cells for 24 h, 48 h and 72 h. (**B**) Comparison of the toxicity effects in HCT116 cells treated with MMNC, 5-Fu for 48 h. (**C**) IC_50_ value of MMNC in HIEC cells for 48 h. (**D**) IC_50_ value of MMNC in Caco-2 cells for 48 h. (**E**) Morphological changes on HCT116 cells after MMNC treatment for 48 h. (**F**) IC_50_ value of 5-Fu in HIEC cells for 48 h. (**G**) Morphological changes on Caco-2 cells after MMNC treatment for 48 h. (**H**) Absorbance of HCT116 cells treated with MMNC at 0.1 and 0.2 μM for 24, 48, and 72 h. Values are shown as the means ± standard, n = 3. *** *p* < 0.001 compared to negative control group.

**Figure 2 ijms-24-15142-f002:**
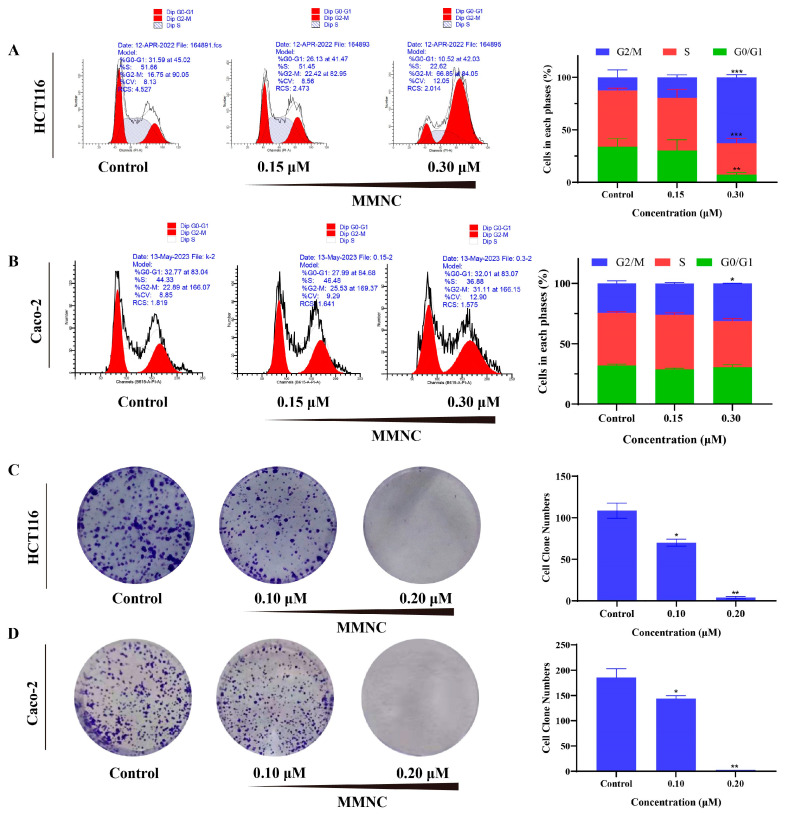
Effects of MMNC on proliferation and cell cycle of HCT116 and Caco-2 cells. (**A**) MMNC arrested HCT116 cells on G2/M phase after treatment for 24 h. (**B**) MMNC arrested Caco-2 cells on G2/M phase after treatment for 24 h. (**C**) MMNC inhibited the proliferation of HCT116 cells after treatment for 7 days. (**D**) MMNC inhibited the proliferation of Caco-2 cells after treatment for 7 days. Values are shown as the means ± standard, n = 3. * *p* < 0.05, ** *p* < 0.01, and *** *p* < 0.001 compared to negative control group.

**Figure 3 ijms-24-15142-f003:**
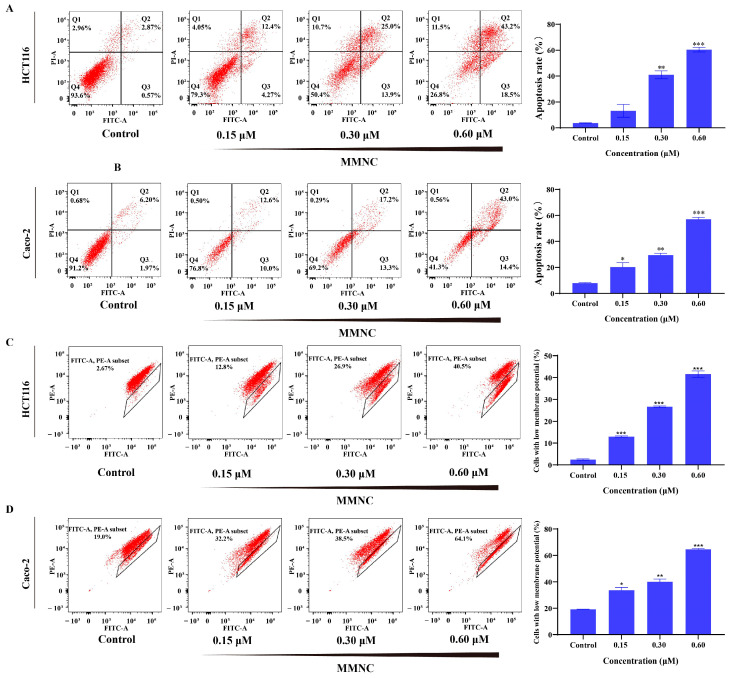
MMNC reduced mitochondrial membrane potential and induced apoptosis. (**A**) MMNC induced apoptosis of HCT116 cells after treatment for 48 h. (**B**) MMNC induced apoptosis of Caco-2 cells after treatment for 48 h. (**C**) The mitochondrial membrane potential of HCT116 cells decreased after treatment with MMNC for 48 h. (**D**) The mitochondrial membrane potential of Caco-2 cells decreased after treatment with MMNC for 48 h. Values are shown as the means ± standard, n = 3. * *p* < 0.05, ** *p* < 0.01, and *** *p* < 0.001 compared to negative control group.

**Figure 4 ijms-24-15142-f004:**
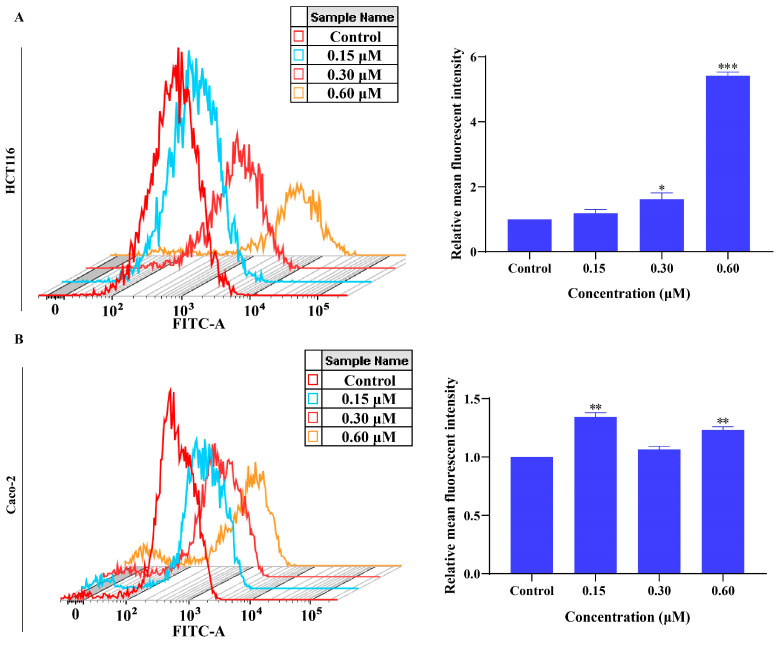
MMNC promoted the production of reactive oxygen species in HCT116 and Caco-2 cells. (**A**) Analysis of ROS induced by MMNC (0, 0.15, 0.3, 0.6 μM) for 48 h using DCFH-DA assay for 48 h in HCT116 cells. (**B**) Analysis of ROS induced by MMNC (0, 0.15, 0.3, 0.6 μM) for 48 h using DCFH-DA assay for 48 h in Caco-2 cells. Values are shown as the means ± standard, n = 3. * *p* < 0.05, ** *p* < 0.01, and *** *p* < 0.001 compared to negative control group.

**Figure 5 ijms-24-15142-f005:**
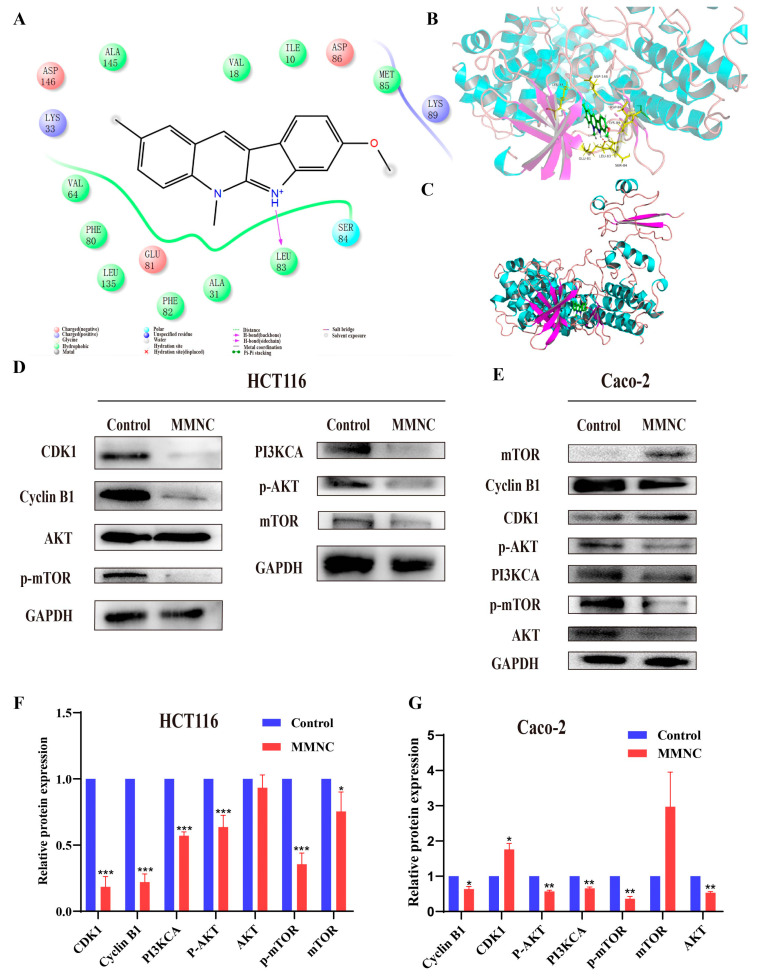
Molecular docking of MMNC and CDK1 protein and the effect of MMNC treatment on PI3K, AKT and mTOR proteins of HCT116 and Caco-2 cells. (**A**) 2D diagram of CDK1 interacting with small-molecule ligand MMNC. (**B**) Detailed interaction of MMNC binding to the CDK1 protein. (**C**) Overall view of MMNC binding to the CDK1 protein. (**D**,**E**) The PI3K/AKT/ mTOR cell signaling pathway-related protein expression were inhibited after treatment with MMNC for 48 h in HCT116 and Caco-2 cells. (**F**,**G**) Analysis of protein expression. Values are shown as the means ± standard, n = 3. * *p* < 0.05, ** *p* < 0.01, and *** *p* < 0.001 compared to negative control group.

**Table 1 ijms-24-15142-t001:** Cytotoxic effects of MMNC on different cancer cell lines for 48 h.

	IC_50_ (μM)					
Compound	HCT116	Caco-2	AGS	PANC-1	SMMC-7721	HIEC
MMNC	0.33	0.51	3.6	18.4	9.7	11.3
Neocryptolepine	6.26	13.54	5.81	>50	19.31	31.37

## Data Availability

Not applicable.

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
