# Peer review of "Biological Evaluation of 8-Methoxy-2,5-dimethyl-5H-indolo[2,3-b] Quinoline as a Potential Antitumor Agent via PI3K/AKT/mTOR Signaling"

_ijms, 2023, doi:10.3390/ijms242015142_

Round 1

Reviewer 1 Report

Dear authors, 

Here is reported a draft focused on the deep analysis of the mechanism of action of a previously synthesized compound, here reported as MMNC. The latter is mentioned as a potential antitumor agent in CRC by interfering with the PI3K/AKT/mTOR signaling.

In general terms the paper is well organized, the aim of the project is clearly outlined, and the results follow the pathway to reach the project goal. Here are some issues that should be fixed prior to publication. 

1. Figures must be added in higher quality format, in this version are not properly readable. 

2. Some typing errors were found throughout the draft, it should be carefully revised.

3. The western blot gels prove the PI3K/AKT/mTOR pathway is affected when cells are treated with MMNC while none proves the CDK1 inhibition. Authors should compare the binding scores of MMNC and a known inhibitor that sits in the same binding pocket. Also, authors should perform in vitro CDK1 inhibition assays to confirm the proposed/supposed target. 

Best regards

Author Response

Dear reviewer #1:

Comment 1: Figures must be added in higher quality format, in this version are not properly readable.

Response 1: Thank you for your sincere advice. We have replaced them with higher-resolution figures in the revised manuscript.

Comment 2: Some typing errors were found throughout the draft, it should be carefully revised.

Response 2: We appreciate the reviewer’s comments.. We have checked the draft carefully and corrected these typing errors.

Comment 3: The western blot gels prove the PI3K/AKT/mTOR pathway is affected when cells are treated with MMNC while none proves the CDK1 inhibition. Authors should compare the binding scores of MMNC and a known inhibitor that sits in the same binding pocket. Also, authors should perform in vitro CDK1 inhibition assays to confirm the proposed/supposed target.

Response 3: Thank you for your kind suggestion. Firstly, in Figure 5D, the protein content of CDK1 in HCT116 cells was significantly reduced after the treatment of MMNC. MMNC inhibited the expression of CDK1 protein. In Figure 5E, the protein content of CDK1 in Caco-2 cells was increased after MMNC treatment. However, the expression of CyclinB1 protein decreased, so the function of CDK1-CyclinB1 cell cycle checkpoint could not be played, resulting in cycle arrest. In our further experiments, we will confirm the target of MMNC and use a known inhibitor for control. The test of CDK1 enzyme activity in vitro will be performed.

Special thanks to you for your valuable comments.

Reviewer 2 Report

The manuscript entitled "Biological Evaluation of the Neocryptolepine Derivative MMNC as a Potential Antitumor Agent via PI3K/AKT/mTOR Signaling" reports the cytotoxic effect of the 8-methoxy-2,5-dimethyl-5H-indolo[2,3-b]quinoline (MMNC) against two cancer cell lines and its underlying mechanisms. The effect appears promising, and the work has been nicely executed, but a few point remain.

Comments:

1 - There are a few typos and grammar issues that need checking and correction.

2 - There is a mismatch between the caption of Table 1 and its information. Please make the necessary adjustments.

3 - On page 2, line 86, "cell morphology crumpled and died" is not understandable. Please rephrase the sentence for clarity.

4 - On page 3, line 90, "exhibited excellent antitumor activity and low toxicity" should be rephrased. Additionally, please clarify what the authors mean by "low toxicity."

5 - In this line, has the compound been tested against non-cancer cell lines? Please address this point on the text.

6 - In section 2.4, it is suggested that a related compound intercalates with DNA, affecting important enzymes. Do the authors think the effect is similar for the current cell lines? From the text, it appears that the effect occurs through interaction with CDK1. Can both mechanisms occur simultaneously? Please provide a more detailed discussion of this point.

7 - In the case of the Caco-2 cell lines, the Western blot revealed different protein levels compared to the HCT116 cell lines; however, this is not explained. Please provide an explanation for these differences.

There are a few typos and grammar issues that need checking and correction.

Author Response

Dear reviewer #2:

Comment 1: There are a few typos and grammar issues that need checking and correction.

Response 1: We appreciate the reviewer’s comments. As with reviewer 1's comment, we have carried out a detailed examination of the manuscript, corrected the grammatical errors and the revised content was shown in red font in the revised manuscript.

Comment 2: There is a mismatch between the caption of Table 1 and its information. Please make the necessary adjustments.

Response 2: We acknowledge the reviewer’s comments and suggestions very much. We have changed the caption of Table 1 to “Cytotoxic effects of MMNC on different cancer cell lines for 48 h.”

Comment 3: On page 2, line 86, "cell morphology crumpled and died" is not understandable. Please rephrase the sentence for clarity.

Response 3: Thank you for your guidance. We have revised “cell morphology crumpled and died” to “cell contracted obviously, had weak adhesion, and became round and loosely ar-ranged” in the revised manuscript.

Comment 4: On page 3, line 90, "exhibited excellent antitumor activity and low toxicity" should be rephrased. Additionally, please clarify what the authors mean by "low toxicity."

Response 4: Thanks for your good instruction. We have rephrased the sentence “exhibited excellent antitumor activity and low toxicity” in the revised manuscript. “low toxicity” means that MMNC behaved excellent cytotoxic activity in the colorectal cancer HCT116 and Caco-2 cells and the cytotoxicity in human intestinal epithelial HIEC cells was low.

Comment 5: In this line, has the compound been tested against non-cancer cell lines? Please address this point on the text.

Response 5: Thank you for your instructive advice. We have tested the cytotoxicity of MMNC on non-cancer cells, which was also described in the revised manuscript. the cytotoxicity in human intestinal epithelial HIEC cells was low. The IC50 of MMNC in human intestinal epithelial HIEC cells was 11.27 μM.

Comment 6: In section 2.4, it is suggested that a related compound intercalates with DNA, affecting important enzymes. Do the authors think the effect is similar for the current cell lines? From the text, it appears that the effect occurs through interaction with CDK1. Can both mechanisms occur simultaneously? Please provide a more detailed discussion of this point.

Response 6: We appreciate the reviewer’s suggestion. MMNC could exert cytotoxic effects by influencing the activity of topoisomerase II and inserting DNA. At the protein level, DNA damage may cause changes in key proteins of PI3K/AKT/mTOR cell signaling pathway, which in turn leads to changes in the content of CDK1, a cell cycle dependent protein, thus playing a cytotoxic role. In this paper, we detected the changes of PI3K/AKT/mTOR cell signaling pathway and the cell cycle dependent protein CDK1. Therefore, in order to further find the target of MMNC, it is necessary to further detect the interaction between MMNC and topoisomerase II and whether it plays an cytotoxic activity by inserting DNA in the following further study.

Comment 7: In the case of the Caco-2 cell lines, the Western blot revealed different protein levels compared to the HCT116 cell lines; however, this is not explained. Please provide an explanation for these differences.

Response 7: We acknowledge the reviewer’s comments and suggestions very much, which are valuable in improving the quality of our manuscript. We have provided a detailed discussion in the Results and Discussion 2.8 of the revised manuscript. After treatment of HCT116 and Caco-2 cells with MMNC, the expression levels of CDK1 and mTOR proteins changed inconsistent. On the one hand, it may be that CDK1 and CyclinB1 protein can play a role in cell cycle regulation only after binding, and mTOR protein can only play an active role after phosphorylation. On the other hand, it may be due to the heterogeneity between tumor cells that results in this difference.

Special thanks to you for your valuable comments.

Reviewer 3 Report

In this article the cytotoxicity of 8-methoxy-2,5-dimethyl-5H-indolo[2,3-b]quinoline (MMNC) in HCT116 and Caco-2 colorectal cells was evaluated. This derivative blocks the cell cycle in the G2/M phase, decreases the cell mitochondrial membrane potential and induces apoptosis. In addition, the results of Western blot experiments suggest that MMNC exerts cytotoxicity by inhibiting the expression of PI3K/AKT/mTOR signaling pathway-related proteins. The manuscript is not well organized and it is not deserve to be published in the present form:

The list of the authors is not complete.

The choice and evaluation of MMNC was not explained.

There are several problems in the organization of the manuscript. All sentence from line 217 to 230 should be reported in the introduction section

The title of table 1 was wrong. Table 1 reported the antiproliferative activity of MMNC agaist a panel of camcer cell lines. In this table Caco-2 cells was not included.

The two intermediated employed for the preparation of final compound were not characterized as 1H NMR spectra.

PI3K and mTOR  inhibition should be evaluated by an in vitro kinase enzymatic assay.

Docking studies are confusing. Figure 5a reported the interaction of MMNC with CDK1. Docking studies should be performed on PI3K and mTOR.

Author Response

Dear reviewer #3:

Comment 1: The list of the authors is not complete.

Response 1: Thanks for your kind suggestion.  We have corrected it in the revised manuscript.

Comment 2: The choice and evaluation of MMNC was not explained.

Response 2: Special thanks to you for your good suggestion. In the Results and Discussion 2.2 section, we have added relevant statements explaining the reasons for choosing MMNC for evaluation in revised manuscript. By evaluating the bioactivity of a series of neocryptolepine derivatives, we found that MMNC showed selective cytotoxicity to colorectal cancer HCT116 and Caco-2 cells, so MMNC was further tested and evaluated for anticancer activity.

Comment 3: There are several problems in the organization of the manuscript. All sentence from line 217 to 230 should be reported in the introduction section.

Response 3: According to your kind suggestions. We have transferred the sentence from line 217 to 230 to the introduction section in the revised manuscript.

Comment 4: The title of table 1 was wrong. Table 1 reported the antiproliferative activity of MMNC agaist a panel of camcer cell lines. In this table Caco-2 cells was not included.

Response 4: Special thanks to you for your good suggestion. We have modified the title of Table 1 to "Cytotoxic effects of MMNC on different cancer cell lines for 48 h". In the revised Table 1, We increased the IC50 of MMNC against colorectal cancer Caco-2 cells for 48 h.

Comment 5: The two intermediated employed for the preparation of final compound were not characterized as 1H NMR spectra.

Response 5: Special thanks to you for your good suggestion. According to reviewer 4's comments, we have deleted the synthesis section, the 1H NMR spectra of the two intermediates and the synthesis route of MMNC can be found from our previous article [1].

Comment 6: PI3K and mTOR  inhibition should be evaluated by an in vitro kinase enzymatic assay.

Response 6: Thanks to you for constructive comments and suggestions. In molecular docking study, MMNC had the good binding ability with CDK1 protein, and the results of molecular docking between PI3K and mTOR and MMNC can be found in our Supplementary Material Table S1. In the following further study, we will evaluate the kinase activity of MMNC with PI3K,mTOR and CDK1 in vitro.

Comment 7: Docking studies are confusing. Figure 5a reported the interaction of MMNC with CDK1. Docking studies should be performed on PI3K and mTOR.

Response 7: Thanks to you for kind suggestions. In molecular docking, MMNC and PI3K,mTOR,CDK1 docking scores are -7.543, -7.450, -8.834 respectively. According to the results of molecular docking, MMNC has the better binding ability with CDK1 protein, so we made the molecular docking diagram of MMNC and CDK1 protein. The docking results of MMNC with PI3K and mTOR can be found in Supplementary Material Table S1.

Special thanks to you for your valuable comments.

Reference

  1. Zhu, J.K., J.M. Gao, C.J. Yang, X.F. Shang, Z.M. Zhao, R.K. Lawoe, R. Zhou, Y. Sun, X.D. Yin, and Y.Q. Liu, Design, Synthesis, and Antifungal Evaluation of Neocryptolepine Derivatives against Phytopathogenic Fungi. J Agric Food Chem, 2020. 68(8): p. 2306-2315.

Reviewer 4 Report

In the manuscript titled “Biological evaluation of the neocryptolepine derivative MMNC as a potential antitumor agent by PI3K/AKT/mTOR signaling” the authors report the biological study on a single derivative of neocryptolepine.

My comments are:

1. In the title it is better not use the acronym MMNC before to have explained the means.

2. At row 5 it is reported “…Peng Chen1 and*” it is not clear if after “and”, it is missing an author.

3. At rows 48-50 it is written: “In our previous study, a series of neocryptolepine derivatives were obtained, of which compound 44 (MMNC) had excellent activity”, but to consider excellent the bioactivity, GI50 should have  nM values. In addition, the reference is missing.  In literature it is published an article (not by the authors of this manuscript), in which are reported a series of neocryptolepine derivatives among them the cited compound MMNC, which  strangely has the same number 44 (Jia-Kai Zhu, Jian-Mei Gao, Cheng-Jie Yang, Xiao-Fei Shang, Zhong-Min Zhao, Raymond Kobla Lawoe, Rui Zhou, Yu Sun, Xiao-Dan Yin, and Ying-Qian Liu*, Design, Synthesis, and Antifungal Evaluation of Neocryptolepine Derivatives against Phytopathogenic Fungi. J. Agric. Food Chem. 2020, 68, 2306−2315. DOI: 10.1021/acs.jafc.9b06793).

4. In results and discussion it is unnecessary report paragraph 2.1 because the synthesis and characterization was already reported in the article cited at the point 3. It is enough to cite the article.

5. The caption of Table 1 is wrong. Table 1 needs to be insert in the paragraph 2.2 and not before. This table is very poor, because the authors should have insert the bioactivity of natural neocryptolepine and of a known antitumor compound acting with a similar mechanism of action, in order to compare the bioactivity.

6. At rows 111-113 it is reported that the neocryptolepine  is able to intercalate  the DNA and interferes with Topo II activity blocking cells in G2/M phase as well as MMNC shows the same behavior  as expected due to the minimal structural difference between the two molecules. Why the authors have not also performed docking calculation on the more probable target Topo II?

7. Molecular docking on CDK1 protein gives some nonspecific hydrophobic interactions and only an H-bond between N-H of tetracyclic moiety and Lys83. The same interactions were reported for an isosters (called CFNC) in the previous article published from the same authors (European Journal of Pharmacology 938 (2023) 175408. DOI: 10.1016/j.ejphar.2022.175408). For this reason, the substitution in the tetracyclic unit of neocryptolepine doesn’t modulate the interactions with the forecasted target.

8. The resolution of Figure 5 needs to be improved.

9. The discussion is poor.

10. In material and methods, all paragraphs from 3.1 to 3.2.3, for the same reason of point 4 of my revision, need to be removed and it is enough written: “MMNC was prepared as reported in [ref]. (wher[ref] means the number of reference of J. Agric. Food Chem. 2020, 68, 2306−2315. DOI: 10.1021/acs.jafc.9b06793).

11.   Paragraph 3.10 concerning Docking needs to be improved adding further details, in order to allow the reader to repeat the docking in the same conditions.

In conclusion this manuscript gives no novelty or improvement in knowledge on the mechanism of action of neocryptolepine. It is only an application of the same biological assays protocols, as already very recently published from the same authors, on an isosteric derivative of MMNC (called CFNC) on gastric cell lines instead on colon cell lines. In fact the molecule of  already published  study differs only for the substitution of a chlorine atom in the place of a methyl group and for the substitution of a fluorine atom in the place of a methoxy unit (See: European Journal of Pharmacology 938 (2023) 175408. DOI: 10.1016/j.ejphar.2022.175408). The authors have should done a single article containing the biological study on the whole series of neocriptolepine derivatives in order to compare the effect of various substitutions and to obtain a Structure Activity Relationship (SAR). This manuscript overlaps the same results both in PI3K/AKT mechanism and in docking  of the previous recently published article.

Author Response

Dear reviewer #4:

Comment 1: In the title it is better not use the acronym MMNC before to have explained the means.

Response 1: Special thanks to you for your good suggestion. We have modified the title to “Biological evaluation of 8-methoxy-2,5-dimethyl-5H-indolo[2,3-b] quinoline as a potential antitumor agent by PI3K/AKT/mTOR signaling” in revised manuscript.

Comment 2: At row 5 it is reported “…Peng Chen1 and*” it is not clear if after “and”, it is missing an author.

Response 2: Thanks a lot for the reviewer’s suggestion. There is no author in the end, and we have written an extra "and" in the original manuscript, which we have revised in the revised manuscript.

Comment 3: At rows 48-50 it is written: “In our previous study, a series of neocryptolepine derivatives were obtained, of which compound 44 (MMNC) had excellent activity”, but to consider excellent the bioactivity, GI50 should have  nM values. In addition, the reference is missing.  In literature it is published an article (not by the authors of this manuscript), in which are reported a series of neocryptolepine derivatives among them the cited compound MMNC, which  strangely has the same number 44 (Jia-Kai Zhu, Jian-Mei Gao, Cheng-Jie Yang, Xiao-Fei Shang, Zhong-Min Zhao, Raymond Kobla Lawoe, Rui Zhou, Yu Sun, Xiao-Dan Yin, and Ying-Qian Liu*, Design, Synthesis, and Antifungal Evaluation of Neocryptolepine Derivatives against Phytopathogenic Fungi. J. Agric. Food Chem. 2020, 68, 2306−2315. DOI: 10.1021/acs.jafc.9b06793).

Response 3: Special thanks to you for your valuable suggestion. We have added the IC50 of compound MMNC in colorectal cancer HCT116 cells in the revised manuscript, and the relevant reference was cited. In our previous study, a series of neocryptolepine derivatives were obtained, of which compound 44 (MMNC) had excellent activity with an IC50 of 0.33 μM against colorectal cancer HCT116 cells. In our previous article, only the antifungal tests were performed on the synthetic neocryptolepine derivatives. In this paper, we evaluated the cytotoxicity of the neocryptolepine derivative MMNC and studied the possible mechanism of anti-colorectal cancer. Therefore, the number of compound in the two papers is the same. 

Comment 4: In results and discussion it is unnecessary report paragraph 2.1 because the synthesis and characterization was already reported in the article cited at the point 3. It is enough to cite the article.

Response 4: Thanks for your kind suggestion. We have deleted the content about synthesis in paragraph 2.1 and relevant article was cited. The synthesis route and characterization of MMNC can be found in our previous study [1].

Comment 5: The caption of Table 1 is wrong. Table 1 needs to be insert in the paragraph 2.2 and not before. This table is very poor, because the authors should have insert the bioactivity of natural neocryptolepine and of a known antitumor compound acting with a similar mechanism of action, in order to compare the bioactivity.

Response 5: We truly appreciate the reviewer’s comments. We have modified the caption of Table 1 to "Cytotoxic effects of MMNC on different cancer cell lines for 48 h", and have transferred Table 1 to paragraph 2.2. According to the reviewer's suggestion, we have performed experiments, inserting the biological activity of natural neocryptolepine into Table 1, so as to compare the biological activity between them. The biological activity of the known antitumor compound 5-Fu against colorectal cancer HCT116 cells can be found in Figure 1B, and cytotoxicity against human intestinal epithelial HIEC cells can be found in Figure 1F.

Comment 6: At rows 111-113 it is reported that the neocryptolepine  is able to intercalate  the DNA and interferes with Topo II activity blocking cells in G2/M phase as well as MMNC shows the same behavior  as expected due to the minimal structural difference between the two molecules. Why the authors have not also performed docking calculation on the more probable target Topo II?

Response 6: Special thanks to you for your kind suggestion. We did not perform the docking calculation between MMNC and topoisomerase in the molecular docking, because we did not find experimental results of interaction between MMNC and topoisomerase in the pharmacodynamic evaluation. Our experimental results showed that MMNC could induce apoptosis of colorectal cancer HCT116 and Caco-2 cells. Moreover, the cell cycle of HCT116 and Caco-2 cells can be blocked in G2/M phase. Therefore, we selected proteins related to cell cycle, apoptosis and cell signaling pathways for molecular docking studies. We will further evaluate the interaction between MMNC and topoisomerase through molecular docking, enzyme activity test and MST experiments in the following further studies.

Comment 7: Molecular docking on CDK1 protein gives some nonspecific hydrophobic interactions and only an H-bond between N-H of tetracyclic moiety and Lys83. The same interactions were reported for an isosters (called CFNC) in the previous article published from the same authors (European Journal of Pharmacology 938 (2023) 175408. DOI: 10.1016/j.ejphar.2022.175408). For this reason, the substitution in the tetracyclic unit of neocryptolepine doesn’t modulate the interactions with the forecasted target.

Response 7: Thanks to you for constructive comments and suggestions. In molecular docking, MMNC and CDK1 have the good binding ability, and in order to better demonstrate the interaction between MMNC and CDK1, we drew the interaction diagram of MMNC and CDK1 proteins. MMNC and the compound CFNC we reported may have similar interactions with CDK1, the imino H of the compound acts as a hydrogen bond donor to form a stable hydrogen bond interaction with the carbonyl oxygen atom on the 83 leucine (LEU) skeleton on the acceptor. Since aspartic acid (ASP) in number 146 and 86 and glutamic acid (GLU) in number 81 are acidic amino acids, they lose H in the solvent, resulting in a strong negative electric interaction with the ligand small molecule, while tryptophan in number 84 has a strong polar interaction with the ligand small molecule. Lysines 33 and 89 (LYS) are basic amino acids that are positively charged and interact with ligands in solvents. The ligand small molecule has some atoms exposed to the solvent.

Comment 8: The resolution of Figure 5 needs to be improved.

Response 8: Thanks to you for your valuable suggestions. We have replaced Figure 5 with a Figure of higher resolution.

Comment 9: The discussion is poor.

Response 9: We thank the reviewer’s valuable suggestion. We have revised and improved the discussion of the paper in the Result and Discussion section, which is mainly in Result and discussion 2.2, 2.2, 2.8. The revised content is shown in red font in the revised manuscript.

Comment 10: In material and methods, all paragraphs from 3.1 to 3.2.3, for the same reason of point 4 of my revision, need to be removed and it is enough written: “MMNC was prepared as reported in [ref]. (wher[ref] means the number of reference of J. Agric. Food Chem. 2020, 68, 2306−2315. DOI: 10.1021/acs.jafc.9b06793).

Response 10: We sincerely appreciate the reviewer’s comments. We have removed the sentence from 3.1 to 3.2.3 and modified it to “MMNC was prepared as reported in our previous study” in revised manuscript.

Comment 11: Paragraph 3.10 concerning Docking needs to be improved adding further details, in order to allow the reader to repeat the docking in the same conditions.

Response 11: Thank you for your instructive advice. We have added the further details in revised manuscript. The center of the small molecule binding to the protein was used as the docking site of CFNC and the proteins for analysis. All structures of proteins can be obtained from PDB database. The PDB ID, resolution and organisms can be found in Supplementary materials.

Comment 12: In conclusion this manuscript gives no novelty or improvement in knowledge on the mechanism of action of neocryptolepine. It is only an application of the same biological assays protocols, as already very recently published from the same authors, on an isosteric derivative of MMNC (called CFNC) on gastric cell lines instead on colon cell lines. In fact the molecule of  already published  study differs only for the substitution of a chlorine atom in the place of a methyl group and for the substitution of a fluorine atom in the place of a methoxy unit (See: European Journal of Pharmacology 938 (2023) 175408. DOI: 10.1016/j.ejphar.2022.175408). The authors have should done a single article containing the biological study on the whole series of neocriptolepine derivatives in order to compare the effect of various substitutions and to obtain a Structure Activity Relationship (SAR). This manuscript overlaps the same results both in PI3K/AKT mechanism and in docking  of the previous recently published article.

Response 12: We sincerely appreciate the reviewer’s comments. In our previous studies, we first synthesized a series of neocryptolepine derivatives and evaluated the antifungal activity of these compounds. In the following study, we found that the derivative CFNC had selective cytotoxicity to gastric cancer AGS cells, and the possible cytotoxicity mechanism was studied. In this paper, we expanded the anticancer profile of neocryptolepine derivatives, and unexpectedly found that the neocryptolepine derivative MMNC has good selective cytotoxicity against colorectal cancer HCT116 and Caco-2 cells, and performed preliminary study on the mechanism of its anticancer activity. In the next study, we will further study the target of the derivatives and the detailed molecular mechanism.

Special thanks to you for your valuable comments.

Reference

  1. Zhu, J.K., J.M. Gao, C.J. Yang, X.F. Shang, Z.M. Zhao, R.K. Lawoe, R. Zhou, Y. Sun, X.D. Yin, and Y.Q. Liu, Design, Synthesis, and Antifungal Evaluation of Neocryptolepine Derivatives against Phytopathogenic Fungi. J Agric Food Chem, 2020. 68(8): p. 2306-2315.

Round 2

Author Response

Dear reviewer #1:

Response : Thank you for your sincere advice. Thank you very much for reviewing our manuscript.

Special thanks to you for your valuable comments.

Reviewer 3 Report

In vitro PI3K,mTOR and CDK1 assays were fundamental for the rational of the manuscript. IC50 values should be calculated by in vitro assay. 

Author Response

Dear reviewer #3:

Comment 1: In vitro PI3K,mTOR and CDK1 assays were fundamental for the rational of the manuscript. IC50 values should be calculated by in vitro assay.

Response : Thank you for your sincere advice. In molecular docking results, MMNC had the good interaction with CDK1 protein. In western blot experiment, MMNC reduced the expression of CDK1 protein in HCT116 cells. Moreover, western blot results also proved that MMNC could exert cytotoxic effects by regulating the PI3K/AKT/mTOR cell signaling pathway in colorectal cancer HCT116 and Caco-2 cell. Therefore, the conclusion in this paper is that MMNC may inhibit the proliferation of colorectal cancer HCT116 and Caco-2 cells by affecting the PI3K/AKT/mTOR cell signaling pathway. The current experiments had been confirmed the conclusion of the paper. In the process of further molecular mechanism research, we will perform enzyme activity experiments between MMNC and CDK1, topoisomerase. Thus, the target protein of MMNC was identified, and the mechanism of MMNC inhibiting the proliferation of colorectal cancer cells was further revealed. We have added relevant statements for discussion in blue font in Results and Discussion 2.8 of the revised manuscript.

Special thanks to you for your valuable comments.

Reviewer 4 Report

The authors have improved the manuscript following my indications

Author Response

Dear reviewer #4:

Comment 1: The authors have improved the manuscript following my indications.

Response 1: Thanks a lot to reviewers for the review and suggestion.

Special thanks to you for your valuable comments.